# PUTTING IT ALL INTO CONTEXT: SIMPLIFYING AGENTS WITH LCLMS

## ABSTRACT

Recent advances in language model (LM) agents have demonstrated significant potential for automating complex real-world tasks. To make progress on these difficult tasks, LM agent architectures have become increasingly complex, often incorporating multi-step retrieval tools, multiple agents, and scaffolding adapted to the underlying LM. In this work, we investigate whether all of this complexity is necessary, or if parts of these scaffolds can be removed on challenging tasks like SWE-bench. We show that in the case of SWE-bench, simply putting the entire environment into the context of a long context language model (LCLM) and properly prompting the model makes it competitive with carefully tuned, complex agent scaffolds. We show that a Gemini-1.5-Pro model without any scaffolding or tools achieves 38% on SWE-Bench-Verified, comparable with approaches using carefully tuned agent scaffolds (32%). While the unscaffolded approach with Gemini-1.5-Pro falls short of the strongest agentic architectures, we demonstrate that the more capable Gemini-2.5-Pro using the same unscaffolded approach directly attains a 50.8% solve rate. Additionally, a two-stage approach combining Gemini-1.5-Pro with Claude-3.7 achieves a competitive 48.6% solve rate. These results suggest that LCLMs can enable a more monolithic design, reducing reliance on exploration scaffolds in fully observable regimes.

## 1  INTRODUCTION

Recent years have witnessed remarkable progress in language model (LM) agents, demonstrating their capability to perform complex real-world tasks autonomously (Yao et al., 2023; Shen et al., 2023). These systems leverage LMs to propose actions that can be executed through API calls in various environments, enabling applications ranging from repository-level software engineering (Wang et al., 2025; Yang et al., 2024b) to robot control (Ichter et al., 2023; Driess et al., 2023) and scientific experimentation (Boiko et al., 2023; Bran et al., 2023).

Typically, LM agents interact with the environment to gather information, reason, and then execute actions based on this information to achieve pre-specified objectives. Effective exploration of the environment forms the foundation of these systems, yet remains a major challenge. Recent works address this by developing agentic scaffoldings with well-designed tools (Schick et al., 2023; Yang et al., 2024b), external memory (Park et al., 2023b), and specialized pipelines (Shinn et al., 2023; Xia et al., 2024) for effective information gathering. While these methods have demonstrated promising results, they heavily rely on human-engineered scaffoldings tailored to specific LMs or tasks. This reliance stems primarily from the standard assumption that agents operate under partial observability, necessitating active collection of observations to incrementally build their understanding of the environment. While this assumption is true in some cases, such as a VLM agent operating a robot in an unknown environment or a web navigation agent visiting a new website, the environment is *fully observable* in many agentic tasks, such as SWE-bench, where the full repository is accessible to the agent at the start of the task.

Our work asks whether many of these LM agent tasks, such as SWE-bench, can be addressed with much simpler architectures. In other words: *how much would we lose if we replaced complex information gathering processes with a single, long-context LM?* Understanding this question has important implications for our understanding of agent design. Although a pure LCLM-based approach is more expensive than an agentic system, if a single long-context model is competitive with complex scaffolding methods, further improvements in the capability and cost of LCLMs may lead to a simpler

Figure 1: **Overview**. (Left) Traditional agent design treats the environment as partially observable and curates elaborate scaffolding (such as tools and execution pipelines) upon LMs to collect the necessary information for solving the task. (Right) In contrast, we leverage the increasingly powerful capabilities of long-context LMs to develop *state-in-context* agents that eliminate the need for complex scaffolding. These agents achieve full observability by maintaining the entire environment state within the context of LMs, turning open-ended agentic tasks into direct, close-ended tasks where LMs excel.

and more monolithic future, as a single LCLM system would be simpler to train end-to-end. Such a move towards a more monolithic system has significant precedent, mirroring past developments in computer vision (Krizhevsky et al., 2012), statistical machine translation (Bahdanau et al., 2015), and NLP (Brown et al., 2020), where complex and specialized pipelines were replaced by monolithic neural networks.

We take the first steps to answering this broad question by studying SWE-bench – a prototypical task where complex agent scaffolds (Yang et al., 2024b; Wang et al., 2025) are used in the existing state of the art, but there is no inherent need for active information gathering. Specifically, we test whether a scaffolding-free, long-context approach with minimal tools can replace exploration scaffolds in fully observable settings, effectively reducing agent design to a prompting task. Using SWE-Bench-Verified (Chowdhury et al., 2024) as a testbed, we identify a few simple but effective prompting tricks for LCLMs and show that this approach outperforms scaffolding-based baselines when both approaches use the same LLM. Nevertheless, a performance gap remains relative to state-of-the-art agentic systems, primarily due to the relatively weaker coding capabilities of LCLMs compared to state-of-the-art LMs. To address this gap, we demonstrate that a straightforward hybrid approach, combining both the long-context processing capabilities of LCLMs and superior problem-solving capabilities of short-context LMs, substantially closes this performance gap. Our aim is to establish a simple, general but often-overlooked baseline that leverages improving LCLM capability and to map the trade-offs that inform future agent design. While recasting the task as prompting can increase compute, the higher per-query cost of LCLM inference can be amortized via caching and is likely to decline as LCLM efficiency improves.

Overall, our contributions include:

- We study fully observable agent tasks as a proof of concept, showing that a single long-context pass can largely replace exploration scaffolds and outperform more complex baselines by 3–6% without additional scaffolding or tools.

- We identify a collection of simple prompting techniques that take a Gemini-1.5-Pro based system for SWE-bench-Verified from 9% to 32% without patch validation.

- Our framework demonstrates competitive performance with the leading agents. Specifically, Gemini-1.5-Pro achieves a 48.6% solve rate when utilizing our two-stage modification approach with Claude-3.7, while Gemini-2.5-Pro directly attains a 50.8% solve rate in a single step.

## 2 RELATED WORK

**LM agents for software engineering** With the rapid advancement of autonomous LM-based agents (Yao et al., 2023; Shinn et al., 2023; Schick et al., 2023), there has been growing research interest in developing LM agents for solving repository-level software engineering tasks on benchmarks like SWE-Bench (Jimenez et al., 2024). Recent approaches to address this challenging task can be categorized into agentic frameworks and non-agentic pipelines. The former treats LMs as autonomous agents that iteratively interact with the code environment (Gauthier, 2024; ope, 2024). SWE-Agent (Yang et al., 2024a) pioneered a custom agent-computer interface (ACI) enabling LM agents to interact with repository environments through defined actions. Moatless (moa, 2024) and AutoCodeRover (Zhang et al., 2024) enhance localization capabilities through advanced search and retrieval tools, while SpecRover (Ruan et al., 2024) focus on improving agent scaffolding. OpenHands CodeAct (Wang et al., 2025) represents an established open-source project that excels in both tooling and scaffolding, demonstrating competitive results. The alternative approach focuses on developing more specialized execution pipelines tailored to the software engineering tasks. Agentless (Xia et al., 2024) decomposes the task into localization, repair, and patch validation phases with specific scaffolding for each stage. Related works like CodeMonkey (Ehrlich et al., 2025) and CodeStory (Pani, 2024) maintain this structured approach while exploring inference-time scaling to enhance performance. Unlike prior works that curate specialized tools or execution pipelines for LMs to gather information for problem-solving from the environement, our work leverages LCLMs to directly process the entire environment state, demonstrating performance comparable to these heavily engineered scaffolding methods.

**Long-context LMs** Advances in model architecture (Gu & Dao, 2024; Sun et al., 2024) and infrastructure (Dao et al., 2022; Rajbhandari et al., 2020) have enabled LCLMs with extended context windows and enhanced capabilities. Empirical evidence from Lee et al. (2024); Li et al. (2024) indicates that, given adequate computational resources, LCLMs systematically surpass retrieval-augmented generation (RAG) systems in performance, even without task-specific training. Models such as Longformer (Beltagy et al., 2020) and LongT5 (Guo et al., 2022) have demonstrated remarkable efficacy in document summarization tasks. Building on this foundation, our work investigates the capability of LCLMs to supplant traditional memory and information-gathering processes in LM agent scenarios.

## 3 METHOD

Traditional approaches to designing LM agents typically operate under the assumption that the environment is too large to observe directly and rely upon interactive exploration to gather information. This assumption is natural for humans and in internet-based environments where the environment is far too large for any single computer. However, many important applications of agents (e.g. multi-hop QA, software engineering) do not inherently require active information gathering, and in principle, a sufficiently long context model could simply return the answer directly after being prompted with the entire environment in its context.

We contrast the two views in Figure 1, where we compare scaffolding based approaches that use tools and execution loops for information gathering (left) with a direct approach that simply encodes the context as on long prompt (right). Even in tasks as complex as repository-level bugfixing, all files and their relationships are a relatively compact and complete state that is sufficient for tasks like bug fixing within the repository—-e.g., 98% (Maj et al., 2024) of the github repositories can fit into the 2-million-token context window of Gemini (Team et al., 2023). This motivates the question: Can we simply incorporate the entire state directly into the agent's context to enable full observability and eliminate the need for iterative, interactive exploration?

We design such *state-in-context* agents by leveraging the increasingly powerful long-context LMs (LCLMs) that can effectively process millions of context tokens, as illustrated in Figure 1 (right). This eliminates the need for complex scaffolding design for interactive explorations, effectively transforming open-ended agentic tasks into closed-ended ones, settings where LMs excel by leveraging their (long-)context processing abilities. We first illustrate this conceptual idea in Section 3.1, and develop our implementation in Section 3.2.

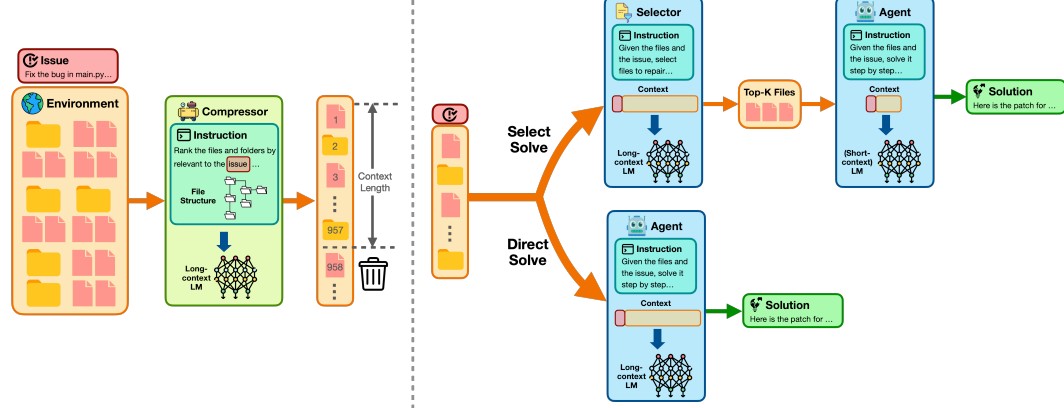

Figure 2: **Instantiation of state-in-context agents for software engineering**. (Left) When the environment state (i.e., code repository) exceeds the context length limit of LCLMs, we apply a simple compression step that ranks files by their relevance and selectively includes files up to the maximum context limit. (Right) We instatiate state-in-context agents in two ways: DIRECTSOLVE directly generates the solution using LCLMs that consume the entire (compressed) state, which are then fed into short-context Language Models (SCLMs) to generate the solutions, leveraging the superior problem-solving capabilities of SCLMs.

## 3.1 DESIGNING STATE-IN-CONTEXT AGENTS WITH LONG-CONTEXT LMS

The standard agent formulation typically operates in *partially observable* settings. In particular, an agent is instantiated to solve a task $p$ (e.g., fixing a bug) within an environment with a state $s$ (e.g., all files and their relations). To accomplish the task, the agent explores the environment by performing a sequence of actions $a_1, a_2, \ldots$ to collect a corresponding sequence of observations $o_1, o_2, \ldots$. Intuitively, the collected set of observations $\cup_{i=1}^{t} o_i$ forms a partial reconstruction of the full state $s_u$, ideally covering the necessary information to complete the task (e.g., buggy code locations). In existing agent frameworks, this interactive exploration is typically implemented through specialized tools, such as a file viewer and code search utility in SWE-Agent (Yang et al., 2024b) or through customized execution pipelines, such as code localization and bug repair in Agentless (Xia et al., 2024) (see Figure 1, left). These approaches require a careful design of agentic scaffoldings tailored to specific tasks that facilitate effective explorations of agents during task executions.

In contrast, we leverage the increasingly powerful long-context processing capabilities of LCLMs to design a *state-in-context* agent, which is an agent that receives either the full state $s$ or a minimally compressed version $\tilde{s} = \mathcal{C}_p(s)$, which retains the task-relevant information (i.e., $s_u \subset \tilde{s}$) while discarding irrelevant details. Because the input state retains the necessary components for solving the task $s_u$, the agent can *explore in context*, using its contextual understanding abilities to extract relevant information and directly produce a solution. This design enables a **minimal pipeline**, reducing the need for hand-crafted, task-specific scaffolding and mitigating the compounding errors that often arise in multi-stage pipelines.

Intuitively, by putting the entire environment state into the model context, our method essentially trades off precision for a better recall of relevant information $s_u$. This approach relies on LCLMs that are sufficiently capable of processing large contexts and effectively retrieving the relevant parts. The bottleneck here is set by the capabilities of LCLMs, which, as we expect, will continue to improve over time. As LCLMs continue to scale, our method may be increasingly favorable–shifting the exploration burden from open-ended interaction to in-context inference and close-ended task-solving where LMs excel.

## 3.2 DEVELOPING STATE-IN-CONTEXT AGENTS FOR SOFTWARE ENGINEERING

In this work, we provide an instantiation of state-in-context agents with LCLMs for software engineering tasks on SWE-Bench (Jimenez et al., 2024). SWE-bench serves as an ideal testbed for our ideas, as it is practically useful and serves as a common agent scaffolding benchmark, but at the same time, there is no inherent need for exploration in SWE-bench.

We aim to develop a simple workflow that minimizes the manual pipeline curation or specialized tool design while maximizing the utility of LCLMs. In particular, the core of our workflow is the use of

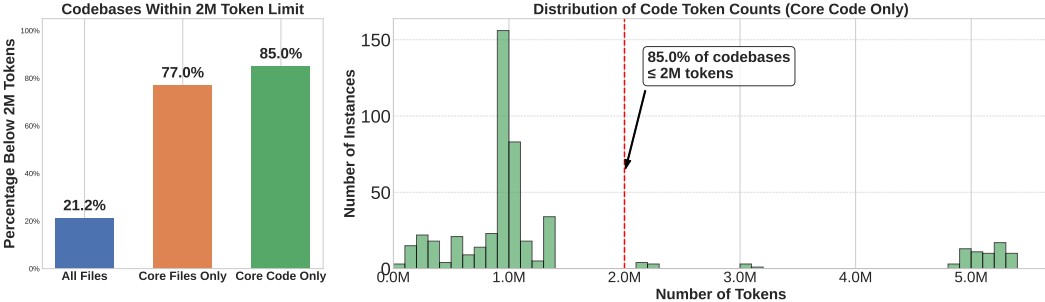

Figure 3: **Codebase token distributions and context window compatibility.** (Left) Percentage of problem instances within the 2M token limit of Gemini-1.5-Pro across three state representation methods: all files, core files only (test and non-python files removed), and core code only (documentation further removed) in SWE-Bench-Verified. (Right) Distribution of token counts for code-only content of instances in SWE-Bench-Verified. The dashed line marks the 2M token threshold, demonstrating that most codebases can fit within current LCLMs context constraints through selective content filtering.

LCLMs that receive the entire code repository and directly outputs the solution (DIRECTSOLVE), analogous to standard zero-shot prompting tasks where LMs have demonstrated strong performance. In cases where the code repositories are too large to fit into the context window, we adopt a simple state compression step with LCLMs based on the repository directory structure to discard obviously irrelevant files. Futhermore, considering that current LCLM capabilities may fall short of the frontier SCLMs, we develop a two-stage approach (SELECTSOLVE) that combine the strength of both to optimize performance. We illustrate our design in Figure 2 and present details below.

**State compression** The context length of current LCLMs may not sufficiently accommodate all code repositories, e.g., the 2-million-token context of Gemini Pro is insufficient for 5 out of 12 repositories (394 out of 500 instances) in SWE-Bench Verified (Chowdhury et al., 2024). This necessitates an additional preprocessing step to properly compress these repositories to fit into the context of current LMs. However, we anticipate that this will become less of an issue as the context length of LCLMs continues to grow.

We implement this compression step using a straightforward ranking-and-filtering approach (see Figure 2, left). In particular, we prompt an LCLM to rank folders and files based solely on the repository structure and the issue statement, producing an ordered list by relevance to the issue (see Section C for the prompt and an example). When some files are omitted from LM's output list, we randomly order them to maintain a complete ranked list. Based on this ranking, we sequentially include as many files into the context as possible until a predefined length threshold (i.e., the maximum context length of the LCLM) is reached. To maximize inclusion, we retain only the core code by removing comments, docstrings, test files, and non-target language files, as these provide limited functional information about the codebase. This approach significantly increases the number of problem instances from SWE-Bench-Verified that can fit into the 2-million-token context window, expanding coverage from 21.2% to 85.0% (see Figure 3). While this approach yields relatively coarse rankings due to limited information from the repository structure, it is sufficient for excluding obviously irrelevant ones and including the target files in the context, guaranteeing a high recall (98.8% on SWE-Bench-Verified in our experiments).

**DIRECTSOLVE method** For code repositories (compressed or otherwise) within the context length limit, our DIRECTSOLVE method directly prompts an LCLM to generate the solution based on the problem statement and the entire (compressed) repository state, as illustrated in Figure 2 (bottom right). We found a combination of two simple but effective prompting tricks to significantly improve performance. Our first prompting technique is *code restatement*, where we prompt the model to re-state relevant code for the task before starting the bug fixing process. This acts as an in-context retrieval mechanism and mitigates the "lost in the middle" issue (Liu et al., 2023) in which LCLMs struggle to extract information from a distant context. The other one we found effective was chain-of-thought prompting (Wei et al., 2023) that generates the code diff after decomposing the solve step into smaller reasoning steps (see Section C). Overall, this approach enables an LCLM to perform localization, code restatement, and repair in a single inference call, requiring only direct prompting without additional handcrafted scaffoldings.

**SELECTSOLVE method** The capabilities of current LCLMs still fall short of short-context counterparts, resulting in pure LCLM-based DIRECTSOLVE agents underperforming compared to carefully designed agentic systems. To address this challenge, we propose a simple two-stage method that leverages both the long-context processing capabilities of LCLMs and the superior problem-solving abilities of SCLMs. The idea is to use an LCLM in the first part where we identify and compress the state and pass this to an SCLM as the problem-solving agent based on the selected files, as illustrated in Figure 2 (top right). In the first step, an LCLM performs single-call localization on the compressed repository, identifying relevant files based on the problem statement (see Section C for the prompt and an example). Although we explored localization at various granularities (functions, methods), file-level localization proved both simpler and more effective. In the second step, a more specialized SCLM (e.g., Claude-3.7 (Anthropic, 2024)) with a limited context performs code repair using only the top K localized files. This approach allows us to combine the strengths of different models while maintaining simplicity.

**Patch format and validation** Both our DIRECTSOLVE and SELECTSOLVE methods generate patches using the Search/Replace edit format introduced in Agentless (Xia et al., 2024), which explicitly specifies both the original code snippets to be replaced and their replacements. This approach enhances precision and reduces hallucination by focusing on targeted edits rather than regenerating entire functions. For patch evaluation and selection, we adopt the same validation methodology from Agentless, which involves generating reproduction tests to verify issue resolution, applying regression tests to minimize functionality disruption, and implementing majority voting on normalized patches to select the most frequently occurring patch as the final submission. While our framework is not primarily focused on inference time scaling, it naturally accommodates computational scaling in a straightforward manner: simply generating more samples in the DIRECTSOLVE method or sampling multiple patches with various localization samples in the SELECTSOLVE method.

## 4 EXPERIMENTS

We describe our experimental setups in Section 4.1. In Section 4.2, we demonstrate the effectiveness of our method with comparable end-to-end results to heavily engineered agentic scaffolding. Section 4.3 presents an extensive ablation study highlighting critical design choices for developing state-in-context agents with LCLMs. Finally, Section 4.4 evaluates the robustness of our approaches across various LCLMs.

### 4.1 EXPERIMENTAL SETUPS

**Benchmark** We evaluate our method using the SWE-bench Verified benchmark (Chowdhury et al., 2024), a high-quality dataset consisting of 500 software engineering problems carefully selected and validated through expert human annotation.

**Implementation details** We used Gemini-1.5-Pro with 2 million token context and Gemini-2.5-Pro with 1 million token context as our LCLM. For SELECTSOLVE, we also evaluated Claude-3.7-Sonnet as the SCLM that demonstrates the specialized performance on coding tasks. We selected the top K = 6 files for SELECTSOLVE, meaning the repair model received content from six localized files to generate patches. Following Agentless (Xia et al., 2024), we used a temperature of 0.8 for sampling different patches and performing patch selection. Across all methods, we generated 8 patches per instance from which a single one is selected by majority voting for evaluation , determined by the batch decoding limit of the Gemini API.

**Baselines** Previous approaches to repository-level software engineering primarily fall into two categories: agentic frameworks and non-agentic scaffolding. Several works employ similar pipelines but focus on scaling up inference-time compute (Ehrlich et al., 2025), which is out of the scope of our comparison–our objective is to demonstrate the effectiveness of our scaffolding-free agent approach, positioning it as a performance-competitive alternative to heavily enginnered agentic scaffoldings. We selected **Agentless** and **CodeAct** as our baselines, representing the state-of-the-art within each method category. We primarily utilized Gemini-1.5-Pro (Team et al., 2023), Gemini-2.5-Pro with google search tool disabled (Google Deepmind, 2025) and Claude-3.7-Sonnet (Anthropic, 2024) to ensure comparability with our methods and included GPT-4o (OpenAI, 2024) with both methods for reference. We describe the details of these baselines below.

Table 1: **Performance across different methods and models on SWE-Bench-Verified**. Our DIRECTSOLVE method, despite its simplicity, outperforms methods with heavily engineered scaffoldings when all methods are instantiated with Gemini-1.5-Pro and Gemini-2.5-Pro. Our SELECTSOLVE method further improves performance over DIRECTSOLVE by leveraging the superior coding capabilities of Claude-3.7-Sonnet, positioning itself between the Agentless and CodeAct approaches when using the same model.

| Approach | Model | Pass@1 | Pass@8 |
|---|---|---|---|
| ***GPT-4o (Reference)*** | | | |
| Agentless | GPT-4o (From cache) | 36.2% | 43.4% |
| CodeAct | GPT-4o | 30.0% | - |
| ***Gemini-1.5-Pro*** | | | |
| Agentless | Gemini-1.5-Pro (Direct transferred) | 11.0% | 13.0% |
| Agentless | Gemini-1.5-Pro (Adapted) | 32.0% | 37.2% |
| CodeAct | Gemini-1.5-Pro | 18.8% | - |
| DIRECTSOLVE | Gemini-1.5-Pro | 38.0% | 46.2% |
| SELECTSOLVE | Gemini-1.5-Pro + Gemini-1.5-Pro | 39.2% | 47.8% |
| ***Claude-3.7-Sonnet*** | | | |
| Agentless | Claude-3.7-Sonnet | 45.2% | 50.8% |
| CodeAct | Claude-3.7-Sonnet (Reported) | 58.0% | - |
| SELECTSOLVE | Gemini-1.5-Pro + Claude-3.7-Sonnet | 48.6% | 59.2% |
| ***Gemini-2.5-Pro*** | | | |
| Agentless | Gemini-2.5-Pro (Adapted) | 47.8% | 54.0% |
| CodeAct | Gemini-2.5-Pro (Reported) | 46.4% | - |
| DIRECTSOLVE | Gemini-2.5-Pro | 50.8% | 60.2% |

- **Agentless** (Xia et al., 2024) was designed specifically for GPT-4o and Claude-3.5-Sonnet, and failed to transfer directly to Gemini models due to compounding parsing errors. We therefore curated the pipeline to ensure correct parsing at each step, and report both pre- and post-curation numbers, labeled as "Direct Transferred" and "Adapted" respectively in Table 1. Moreover, the authors provided intermediate caches of GPT-4o trajectories, allowing us to evaluate them as a completely faithful reproduction. In all Agentless configurations, we sampled 4 instances of localized relevant code following the original work, and generated 2 samples based on each to constitute the 8 patches for selection and evaluation. We followed the same procedure as the original work to select the final patch with majority vote, similar to our method.

- **CodeAct** (Wang et al., 2025) OpenHands CodeAct is an open-source framework that implements an interactive LM-based agent with a well-designed pipeline and toolset. We used the repository at commit `d9926d24` with the default maximum iteration limit of 60, where each iteration consists of an observation and action generation process. Regarding the tool set, we restricted possible tools to basic functionalities, disabling web browsing which is designed for open-ended real-world tasks and could potentially compromise benchmark integrity. For Claude-3.7-Sonnet, we used OpenHands reported results.

**Evaluation metrics** We reported the pass@1 rate, which measures the solve rate of the final selected patch for each method. For our method and the Agentless approach–where we had already sampled 8 potential patches for selection–we additionally reported the pass@8 rate, which measures the rate of at least one patch solving the issue among the 8 samples.

### 4.2 END-TO-END RESULTS ON SWE-BENCH-VERIFIED

**DIRECTSOLVE with LCLMs outperforms agent scaffoldings in matched comparisons** Table 1 shows the end-to-end performance of our methods compared to baselines. When all methods are instantiated with Gemini-1.5-Pro, our DIRECTSOLVE method, despite its simplicity, outperforms the best baseline (Agentless) by 6% on pass@1 with statistical significance. Similarly, implementing Gemini-2.5-Pro yields a statistically significant 6.2% improvement pass@8 metrics. We assess statistical significance using one-sided McNemar tests, with detailed p-values provided in Section A.2. Our DIRECTSOLVE method essentially zero-shot prompts an LCLM to reason and solve problems based on the entire (or compressed) environment state, indicating that current LCLMs can effectively function as scaffolding-free agents. Surprisingly, our DIRECTSOLVE method performs comparably

| Ablation | P@1 | P@8 |
|---|---|---|
| DirectSolve method | 32% | 50% |
|   - CoT prompt | 9% | 16% |
|   - relevant code restatement | 28% | 49% |
|   - add file index | 27% | 44% |
|   - rm comments & docstrings | 27% | 41% |

Table 2: Ablation study on key components of our DIRECTSOLVE method. Results are on a random subset of 100 instances from SWE-Bench-Verified.

| Method | K=3 | K=6 | K=10 |
|---|---|---|---|
| Agentless | | | |
|   prompting-based | 138 | 116 | – |
|   embedding-based | 165 | 119 | 102 |
|   combined | 135 | 88 | 66 |
| LCLM | | | |
|   one-call | 112 | 75 | 56 |
|   one-call (N = 8) | 100 | 62 | 47 |

Table 3: File localization error comparison between Agentless and our LCLM-based file selection in SELECTSOLVE on the whole SWE-Bench-Verified dataset.

with our SELECTSOLVE method when instantiated with the same Gemini-1.5-Pro model (with only a 1.2% lower pass@1 rate), indicating the promise of minimizing agentic workflow when LCLM capabilities continue to improve.

**SELECTSOLVE effectively leverages the capabilities of SCLMs** When our SELECTSOLVE is instantiated with Claude-3.7-Sonnet as the SCLM for generating the patches, the performance improves significantly from 39.2% to 48.6%, highlighting its potential to leverage the advanced problem-solving capabilities of SCLMs. Compared to baselines, SELECTSOLVE outperforms Agentless but falls short of OpenHands CodeAct. This suggests that while our simple method delivers strong performance with minimal engineering effort, specialized tool design tailored to specific models and tasks may still yield additional gains.

**Scaffolding-based baselines do not robustly transfer across models.** Scaffolding-based approaches often rely on specialized tools and execution workflows that are tailored to the behavior of specific models. These specializations—such as prompt designs or tool integrations—are typically optimized for a given model and may not transfer effectively to others. Indeed, we find that for both baselines, direct transfer to Gemini-1.5-Pro without leads to a significant performance drop, with Agentless dropping to 11% and CodeAct to 18.8%. This underscores the limited robustness of standard scaffolding-based approaches across different models, highlighting the necessity of human effort for model-specific adaptations. Consequently, our simplified framework that requires minimal curation may be preferable.

## 4.3 ABLATION STUDY

In this subsection, we systematically analyze various components of our methods to understand their relative contributions to overall performance and identify key factors that enable LCLMs to function effectively as scaffolding-free agents. In all ablation study experiments, we measured performance without applying additional patch validation to specifically assess the intrinsic patch-generation capabilities of LCLMs across various configurations.

**Which design choices matter?** To address this question, we ablate different design choices in our DIRECTSOLVE method to assess the effectiveness of each component. We present the results in Table 2 and detail the discussion below.

- **Prompting Techniques:** We first examine the prompting techniques employed in our approach: chain-of-thought (CoT) prompting, relevant code restatement, and adding file index. Our analysis reveals that **CoT prompting** is crucial for agent performance; removing it leads to a significant performance drop of 23% in pass@1 and 34% in pass@8, underscoring the critical role of reasoning in bug-fixing tasks. Moreover, we find that **relevant code restatement** notably influences pass@1 performance but does not significantly impact pass@8, a metric indicative of solution coverage. This observation suggests that while relevant code restatement may not enahnce the maximum problem-solving capability, it improves the stability and consistency of our method. Finally, we find that the seemingly minor enhancement of **adding file index** information proves important, aligning with the observations in Lee et al. (2024).

- **Repository compression:** We then analyze the effectiveness of **removing comments and docstrings** from the repository to reduce the context length. This reduces the average token count across all instances from approximately 2M to 1.4M and yields an end-to-end performance improvement of 5% in pass@1 and 6% in pass@8. The significant gain through this simple operation

also implies the potential inefficacies of current LCLMs in processing very long-contexts, which we will further analyze in Figure 4.

**How do LCLMs improve localization in the SELECTSOLVE method?** The Agentless framework employs a localization scaffolding that combines direct LM prompting with embedding-based retrieval to localize the specific target files or code lines, whereas our SELECTSOLVE method directly prompts the LCLM to perform localization based on the compressed repository. We evaluate and compare the localization performance of these two approaches by measuring file-level localization errors using top-K file localization with Gemini-1.5-Pro. As shown in Table 3, employing a single LCLM directly can reduce file-level localization error by over 15% compared to the combined scaffolding-based method across various values of K. Moreover, when aggregating localization results from 8 samples using majority voting, the localization error further decreases by approximately 25% to 35%. These results illustrate that utilizing LCLMs not only simplifies the localization procedure but also significantly enhances localization accuracy.

Additional ablations in Section A show that target-file placement and prompt length materially affect LCLM effectiveness, reinforcing the value of relevance-based ordering in the state-compression step and indicating ample headroom for further LCLM development to better exploit long contexts.

### 4.4 ROBUSTNESS TO LCLM CHANGES

Given that the system is designed in the experiments of Gemini-1.5-Pro, we test whether our prompts and results are overfit to Gemini-1.5-Pro by varying the LCLM used in our state-in-context models. Other models with context windows of at least 1 million tokens include: Gemini-1.5-Pro (2M), Gemini-2.5-Pro (1M), Minimax-Text-001 (1M), and Gemini-Flash-2.0 (1M). We randomly sampled 100 instances from SWE-Bench-Verified for this evaluation and did not apply patch validation.

The results are shown in Table 4. Gemini-2.5-Pro consistently achieves the highest performance across all evaluation metrics, attaining a 51% solve rate on a random subset of 100 instances with 8 samples without any patch selection. Notably, DIRECTSOLVE even outperforms SELECTSOLVE methods in the Gemini-2.5-Pro experiments, suggesting that as LCLMs improve, the performance gap between minimal prompting frameworks and scaffolding approaches may further narrow or even reverse.

All methods achieve a nontrivial 20%+ solve rate, suggesting that our experiment setting and prompts are not overfitted to a single LCLM. Minimax-Text-001 (MiniMax et al., 2025), the only open-weight model evaluated, falls short of the closed-source models' performance but still successfully resolves 20% of instances using our SELECTSOLVE method.

Table 4: **State-in-context approaches with alternative LCLMs**. Performance degrades for DirectSolve, but remains relatively high across the available models. Evaluations are conducted on a random subset of 100 instances from SWE-Bench-Verified.

| Approach | Model | p@1 | p@8 |
|---|---|---|---|
| DIRECTSOLVE | Gemini-1.5-Pro | 32% | 50% |
| | Gemini-2.5-Pro | 51% | 64% |
| | Gemini-Flash-2.0 | 24% | 37% |
| | Minimax-Text-001 | 15% | 20% |
| SELECTSOLVE | Gemini-1.5-Pro | 34% | 48% |
| | Gemini-2.5-Pro | 49% | 63% |
| | Gemini-Flash-2.0 | 29% | 46% |
| | Minimax-Text-001 | 20% | 32% |

## 5 DISCUSSION

**Cost analysis** The average cost of Agentless, CodeAct, and our method in Section 4 is $0.25, $0.87, and $2.60, respectively, indicating that LCLM is currently less cost-effective. That said, LM inference costs have fallen sharply—GPT-4–equivalent APIs are down $\sim 1000\times$ over three years (Guido Appenzeller, 2024)—and context windows have expanded $\sim 500\times$ (from 4K to 2M tokens), making monolithic, LCLM-based agents increasingly practical. In real-world use, repeated queries to the same codebase enable KV caching that substantially reduces average inference cost: after the initial pass, the marginal cost is dominated by context-caching tokens (about one-quarter of total), lowering per-instance cost from $2.60 to $\sim$\$0.725. Together, (i) continued inference-cost declines and (ii) KV caching in repeated codebase queries suggest the cost of our simplified method is increasingly acceptable; additional discussion of implications beyond SWE-Bench and general codebase feasibility appears in Section B.

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

# A  ADDITIONAL EXPERIMENTS

In this appendix we extend the ablation analysis from Section 4.3 to examine how target-file placement andcontext length affect model repair performance.

## A.1  ADDITIONAL RESULTS

**How does the target file location impact the agent performance?** The performance of our approach may be bottlenecked by the long-context processing capabilities of LCLMs–they may not sufficiently absorb the information in the provided context and suffer potential issues like "lost in the middle" (Liu et al., 2023). To understand this potential limitation, we conduct a controlled analysis by varying the placement of the target files within the context–

Table 5: Impact of target file location on LCLM DIRECTSOLVE performance with 1M token prompts.

| Metrics | Front | End | Random |
|---------|-------|-----|--------|
| pass@1  | 32%   | 26% | 28%    |
| pass@8  | 50%   | 43% | 41%    |

positioning them at the front, at the end, or randomly within the context–and measure the resulting DIRECTSOLVE performance of Gemini-1.5-Pro. As shown in Table 5, the position of target files significantly impacts the one-step performance, with positioning target files at the front of the prompt yielding the best results. This finding validates our design of ordering files to be included in the context based on their relevance score during the state compression step.

**How does context length impact agent performance?** We further investigate the effect of context length on the performance of state-in-context agents using LCLMs through a controlled analysis. Specifically, we maintain the target files at the front of the context and progressively add remaining files according to their relevance scores—computed during the state compression step—until reaching the specified context length limit. As illustrated in Figure 4, even though the necessary information is consistently present in the prompt, increasing the context length negatively affects solving accuracy. We observe a clear performance degradation as the context length expands from 100K to 1M tokens: pass@8 accuracy decreases from 53% to 47%, and pass@1 accuracy declines from 39% to 31%. This indicates a substantial performance gap for current LCLMs when handling longer contexts in the DIRECTSOLVE approach, a gap potentially alleviated by employing the SELECTSOLVE method. These results also highlight the critical need to enhance LCLMs' capabilities in effectively processing and leveraging longer contexts.

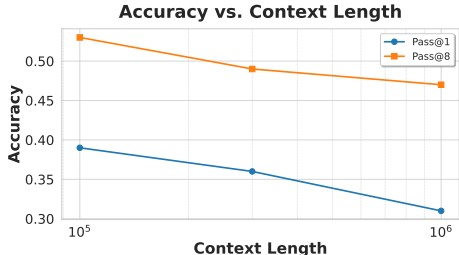

Figure 4: Model performance vs. Context Length for DIRECTSOLVE with target files placed at the front of the prompt.

## A.2  PAIRWISE STATISTICAL SIGNIFICANCE TESTS

We conducted pairwise significance testing on 500 SWE-Bench-Verified instances using one-sided McNemar tests (paired binary outcomes per instance) to evaluate whether our method outperforms the strongest baseline.[1]

**Gemini-1.5-Pro.** Against the strongest baseline (Agentless), DIRECTSOLVE achieves statistical significance at **pass@1** with $p = 0.0018$.

**Gemini-2.5-Pro.** For **pass@1**, the comparison yields $p = 0.07$, which is marginal given a 3% absolute improvement. In contrast, **pass@8** shows clear significance with $p = 0.0008$ against the strongest baseline. We view **pass@8** as the more reliable estimator in this setting because it exhibits lower variance than **pass@1** on fixed-size test sets.

---

[1]We use a one-sided alternative reflecting the directional hypothesis that our method is better.

## B  ADDITIONAL DISCUSSION

**Implications for agents beyond SWE-bench** More broadly, our approach is likely to have implications for scaffolding methods for information gathering, both from the environment as well as learning from their past interaction trajectories (Park et al., 2023a). With CodeAct trajectories averaging 77.7K tokens across SWE-Bench-Verified instances and generative agent simulations reaching approximately 50K tokens (Park et al., 2024), these extensive trajectories fit comfortably within modern LCLMs' context windows. This suggests that LCLMs may also be able to simplify complex memory and retrieval architectures for LM agent interactions.

**General codebase feasibility** We want to emphasize that the repositories in SWE-Bench-Verified represent major collaborative projects and are atypical in their large size. As stated in Maj et al. (2024), less than 2% of GitHub repositories have more than 100K SLOC. This means over 98% of GitHub repositories can probably be directly fitted into the 2M token context window, suggesting that LCLM-based zero-shot approaches may already be practical for simpler use cases.

## C  PROMPT DETAILS

### C.1  DETAILS FOR FILE RANKING

In this section, we provide details about the prompts used in our proposed DIRECTSOLVE and SELECTSOLVE methods presented in Section 3.2.

**File Ranking by relevance** During state compression, we instruct the model to rank files by their relevance to the problem statement given the repository structure. The prompt is as follows:

```
Please look through the following GitHub problem description and Repository structure

Note that you should focus on providing specific files or the lowest subfolder in the

### GitHub Problem Description ###
{problem_statement}

### Repository Structure ###
{structure}

###

Please provide the ranked list with the most relevant item first and the least releva
Ensure that each listed item is directly related to solving the problem described.
The returned list should be separated by new lines and wrapped with '''.
For example:
'''
file1.py
folder2/file3.py
folder4/subfolder5/
folder6/file7.py
'''
```

Through this prompt, the language model produces an ordered list of files and folders, ranked by relevance to the issue statement. Files not explicitly mentioned are randomly appended after all mentioned files. This approach yields an approximated total ranking of all files, enabling us to compress the state to accommodate any context limit.

### C.2  DETAILS AND AN EXAMPLE FOR REPAIR PROMPT

For our DIRECTSOLVE method and the repair stage of SELECTSOLVE, we employ a chain-of-thought (CoT) prompt that guides the model to restate relevant code and generate appropriate repair patches. This prompt provides step-wise instructions with CoT techniques applied to both code localization and repair. While the prompt is extensive and could potentially be simplified in future work, our current implementation uses the following:

You are a senior software engineer tasked with analyzing and resolving a repository

# REPOSITORY STRUCTURE:
--------------------
{file_structure}
--------------------

###########################
# ISSUE DESCRIPTION:
-----------------
{issue}

ANALYSIS INSTRUCTIONS:
--------------------

Your task is to perform the following steps in order:

1. **Chain-of-Thought for Localization**
- Analyze the provided repository structure and issue description to identify the rel
- Explain your reasoning and process for localizing the relevant code.

2. **Restated Relevant Code**
- Provide the exact code snippet that you have identified as relevant to the issue.
- Include a few lines of context before and after the critical section.
- **IMPORTANT:** Enclose this section in a code block using the tag **relevant code**
- If necessary, copy a longer context from the file to ensure that the location where
- If you would like to add the line '
print(x)', you must fully write that out, with all those spaces before the code! Plea
- The format should be as follows:

```relevant code
### path/to/file.py
[Exact code snippet with proper indentation, including sufficient context]
```

3. **Chain-of-Thought for Repairing the Code**
- Explain your reasoning and analysis for repairing the identified issue.
- Describe the necessary modifications, why they are needed, and include any edge cas

4. **Final Patch**
- Provide the final patch using the following exact *SEARCH/REPLACE* format:
 1. The file path.
 2. The start of the search block: <<<<<<< SEARCH
 3. A contiguous chunk of lines to search for in the existing source code.
 4. The dividing line: =======
 5. The lines to replace into the source code.
 6. The end of the replace block: >>>>>>> REPLACE
- **IMPORTANT:** Enclose each final patch in a separate markdown code block using the
- Example format:

```python
### path/to/file.py
<<<<<<< SEARCH
[Original code snippet with proper indentation]
=======
[Replacement code snippet with proper indentation]
>>>>>>> REPLACE
```

864  Requirements:
865  – Focus only on the reported issue.
866  – Provide minimal, precise changes.
867  – Consider error handling and edge cases.
868  – Maintain existing code patterns. If you would like to add the line '
869  print(x)', you must fully write that out, with all those spaces before the code! Plea
870  – Do not reveal internal chain−of−thought details outside of the sections specified a
871
872  Your final output must include these sections in the following order:
873  1. Chain−of−Thought for Localization
874  2. Restated Relevant Code (enclosed in a '''relevant code''' block as specified)
875  3. Chain−of−Thought for Repairing the Code
876  4. Final Patch (each file's modifications enclosed in its own '''python''' code block

877  This prompt elicits a chain-of-thought process for localizing relevant code from files, identifying
878  relevant code segments, analyzing potential repairs, and finally generating a Python diff format patch.
879  For illustration, we provide an example of a problem statement and corresponding model output from
880  the django__django-13516 instance.

881  **Example**
882
883  • **Problem statement**
884
885  flush() on self.stdout/stderr management commands doesn't work.
886  Description
887
888  flush() is notably called during migrate command; it doesn't work, and a long migra
889  Operations to perform:
890   Apply all migrations: myapp
891  Running migrations:
892  Then nothing more, but the migration is being done.
893  Then at the end of the real migration, the rest is flushed:
894   Applying myapp.0002_auto_20200817_1030... OK
895  Expected behavior:
896  Operations to perform:
897   Apply all migrations: myapp
898  Running migrations:
899   Applying myapp.0002_auto_20200817_1030...
900  then work
901  then OK

902  • **Model output**
903
904  1. ∗∗Chain−of−Thought for Localization∗∗
905
906  The issue description states that 'flush()' calls on 'self.stdout' and 'self.stderr
907  Specifically, output is buffered until the end of the command execution, rather tha
908  Looking at 'migrate.py', we see that the 'migration_progress_callback' function wit
909
910  2. ∗∗Restated Relevant Code∗∗
911
912  '''relevant code
913  ### django/core/management/base.py
914  class OutputWrapper(TextIOBase):
915      @property
916      def style_func(self):
917          return self._style_func
       @style_func.setter
       def style_func(self, style_func):
           if style_func and self.isatty():
               self._style_func = style_func

```
            else:
                self._style_func = lambda x: x
        def __init__(self, out, ending='\n'):
            self._out = out
            self.style_func = None
            self.ending = ending
        def __getattr__(self, name):
            return getattr(self._out, name)
        def isatty(self):
            return hasattr(self._out, 'isatty') and self._out.isatty()
        def write(self, msg='', style_func=None, ending=None):
            ending = self.ending if ending is None else ending
            if ending and not msg.endswith(ending):
                msg += ending
            style_func = style_func or self.style_func
            self._out.write(style_func(msg))
```

```relevant code
### django/core/management/commands/migrate.py
        def migration_progress_callback(self, action, migration=None, fake=False):
            if self.verbosity >= 1:
                compute_time = self.verbosity > 1
                if action == "apply_start":
                    if compute_time:
                        self.start = time.monotonic()
                    self.stdout.write("  Applying %s..." % migration, ending="")
                    self.stdout.flush()
```

3. **Chain-of-Thought for Repairing the Code**

The `OutputWrapper` class in `base.py` is missing a `flush` method. As a result, wl

4. **Final Patch**

```python
### django/core/management/base.py
<<<<<<< SEARCH
    def write(self, msg='', style_func=None, ending=None):
        ending = self.ending if ending is None else ending
        if ending and not msg.endswith(ending):
            msg += ending
        style_func = style_func or self.style_func
        self._out.write(style_func(msg))
=======
    def write(self, msg='', style_func=None, ending=None):
        ending = self.ending if ending is None else ending
        if ending and not msg.endswith(ending):
            msg += ending
        style_func = style_func or self.style_func
        self._out.write(style_func(msg))
        self._out.flush()

    def flush(self):
        self._out.flush()

>>>>>>> REPLACE
```

## C.3 DETAILS AND AN EXAMPLE FOR SELECT PROMPT

**Select Prompt.** In the select stage of SELECTSOLVE, we use a prompt to identify the top K most relevant files from the compressed repository. Our prompt is structured as follows:

```
You are a senior software engineer performing a thorough impact analysis. Your task

# REPOSITORY STRUCTURE:
--------------------
{file_structure}
--------------------

############################
# ISSUE DESCRIPTION:
-----------------
{issue}

############################
# ANALYSIS INSTRUCTIONS:
--------------------
Perform the following steps carefully:

1. **Root Cause Analysis**
- Examine the issue description for error patterns and symptoms.
- Trace the flow of data and dependencies across components.
- Identify potential propagation paths of the issue.
- Consider edge cases and failure scenarios.
- Think about related functionalities that could be impacted.

2. **List of Potentially Affected Files**
- Identify all files that may require inspection or modification.
- Include full file paths, listing one file per line.
- Order files by relevance, with the most critical files first.
- Include both directly and indirectly affected files.
- **Err on the side of over-inclusion rather than exclusion.**

**IMPORTANT:** When in doubt, **include** files that might be relevant rather than e

############################
# EXAMPLE RESPONSE:
--------------------

### REASONING:
[Provide a detailed explanation of your analysis process and why certain files may be

### AFFECTED FILES:
src/auth/login.py
src/middleware/auth.py
src/models/user.py
src/api/endpoints/auth.py
config/auth_settings.py
```

Through this prompt, the language model produces an ordered list of relevant files based on the full file contents. We select the first K files and pass them to the repair stage. As an example, in the mwaskom__seaborn-3187 instance, the golden target files were seaborn/utils.py and seaborn/_core/scales.py, which the LCLM correctly identified through chain-of-thought reasoning.

**Example**

• **Problem Statement**

Wrong legend values of large ranges
As of 0.12.1, legends describing large numbers that were created using `ScalarForma
```python
import seaborn as sns
import seaborn.objects as so

penguins = sns.load_dataset("Penguins")
penguins["body_mass_mg"] = penguins["body_mass_g"]*1000
(
    so.Plot(
        penguins, x="bill_length_mm", y="bill_depth_mm",
        color="species", pointsize="body_mass_mg",
    )
    .add(so.Dot())
)
```
The code creates the following plot:
![image](https://user-images.githubusercontent.com/13831112/205512305-778966db-f8d8
which is wrong because `body_mass_mg` is in the order of 1E6. The issue also reprod

I believe the issue stems from not using the offset value of the `ScalarFormatter`
https://github.com/mwaskom/seaborn/blob/ba786bc14eb255f6b4fb7619c8210c5a8016a26f/se
Examining the code of `ScalarFormatter` suggests the issue also depends on the foll
`mpl.rcParams['axes.formatter.useoffset']`
`mpl.rcParams['axes.formatter.offset_threshold']`
However, I did not test it.

The offset value can be safely retrieved from all formatters and based on that it c

- **Model output**

  ### REASONING:

  The issue description points to a problem with legend generation when using ScalarF
  The issue manifests in both `scatterplot` and the `objects` interface using `so.Plo

  The analysis should focus on:

  1. **Scale setup and transformation:** How the `Continuous` scale is set up, includ
  2. **Locator and formatter retrieval:** How the `ScalarFormatter` is being retrieve
  3. **Legend data generation:** How legend entries are created from the scale, parti
  4. **Matplotlib integration:** How seaborn interacts with matplotlib's `ScalarForma
  5. **Affected plot types:** The issue is reproduced with `scatterplot` and `so.Dot
  These include lineplots, histograms, and potentially other distribution or relation

  ### AFFECTED FILES:

  seaborn/_core/scales.py  (Most likely direct cause)
  seaborn/_core/plot.py (Legend generation logic in Plotter)
  seaborn/objects.py (Objects interface using affected scale)
  seaborn/relational.py (Relational plots, specifically scatterplot)
  seaborn/_marks/dot.py (Mark drawing, might need adjustment if scale changes)
  seaborn/distributions.py (Distribution plots that use Continuous scale)
  seaborn/_testing.py (Tests for scales and affected plots will need updating)
  seaborn/regression.py (Regression plots could be affected if using Continuous scale
  seaborn/axisgrid.py (FacetGrid and related plots might use affected scales)
  seaborn/_compat.py (Matplotlib compatibility layer, especially for scaling)
  seaborn/_core/properties.py (Property definitions, might need to handle offset)
  seaborn/utils.py (Utility functions related to scales, legends, and matplotlib inte

```
seaborn/_core/typing.py (Type hints relating to scales and data)
seaborn/__init__.py (Imports and version information)
doc/conf.py (Documentation configuration, especially example generation)

While some files are less likely to be directly involved, they are included for co
Thorough testing across different plot types and parameter combinations will be cru
```

## D USE OF LARGE LANGUAGE MODEL

While the paper's ideas, analysis, and structure were manually drafted by the authors, we employed a LLM as a writing assistant to refine the paper. Specifically, the LLM was used to polish grammar, improve clarity, and adjust wording based on our own words.

