# OpenReview forum: "Putting It All into Context: Simplifying Agents with LCLMs"
_ICLR.cc/2026/Conference — Submitted to ICLR 2026_

### Official Review · Reviewer_aF6w · 2025-10-30

**Soundness:** 2
**Presentation:** 2
**Contribution:** 2
**Rating:** 4
**Confidence:** 4

**Summary:**

This paper shows that dumping the entire codebase into a long-context LM (LCLM) and prompting it directly matches or beats heavily engineered agent pipelines on SWE-bench: Gemini-1.5-Pro scores 38 % and Gemini-2.5-Pro hits 50.8 % without any tools, retrieval, or multi-step scaffolding. The authors compress large repos to fit the context window, apply simple chain-of-thought prompts, and optionally let a stronger short-context model refine the patch. Results demonstrate that full observability plus in-context reasoning turns open-ended software-engineering tasks into closed-ended QA, questioning the need for complex agent frameworks.

**Strengths:**

- The method transfers robustly across Gemini, Claude, and open-sourced models without task-specific tuning.
- The work identifies simple techniques boost real-world system performance, offering practical value to applied practitioners.
- The paper shows strong empirical results, rivaling or surpassing heavily engineered baselines.

**Weaknesses:**

- Missing discussion of recent agentic framework. For instance, CodeAct and Agentess were preprinted in 2024, while there are other recent works in this area which address similar questions.
- Lack of mathematical details for core steps. The key steps of file ranking, compression, and patch selection are described algorithmically but lack precise mathematical formulation or notation.
- Insufficient analysis and experiments.The paper illustrate how method compression enables most SWE-Bench tasks to fit in a 2M-token context, but the approach may not generalize to even larger codebases in practice (most cases in practice). The scalability claims are not demonstrated with out-of-benchmark tasks.

**Questions:**

- Could you please provide more comparison with more recent works?
- It would be better if the author could conduct more experiments on other benchmarks to demonstrate the universality of the conclusion.
- Please give more detailed mathematical formulation or notation about the framework, not just the description of the whole pipeline.

---

### Official Review · Reviewer_dXQ4 · 2025-10-31

**Soundness:** 2
**Presentation:** 2
**Contribution:** 2
**Rating:** 2
**Confidence:** 4

**Summary:**

This paper proposes a new paradigm for intelligent agent design: by directly inputting the entire environment state into a long-context language model (LCLM), it replaces the complex information-gathering scaffolding (such as tool calls and multi-round interactions) in traditional intelligent agents. Experiments on SWE-bench (a software engineering task benchmark) demonstrate that this "state-in-context" approach can match or even surpass the performance of traditional complex agent systems without the need for scaffolding. For example, Gemini-1.5-Pro achieved a 38% success rate on SWE-Bench-Verified, while Gemini-2.5-Pro reached 50.8%.

**Strengths:**

1.Highly Original: Proposed "state-in-context" agents, revolutionizing traditional multi-round interactive agent design.

2.Simple Method: No complex toolchain required, relying solely on prompts and context to complete complex tasks.

3.Excellent Performance: Gemini-2.5-Pro's DIRECTSOLVE method achieved a 50.8% success rate, surpassing most scaffolding systems.

4.Scalability: Supports context compression and multi-model collaboration (e.g., SELECTSOLVE), adapting to diverse models and tasks.

5.Extensive Ablation Experiments: Verified the impact of key factors such as prompting techniques, file ordering, and context length.

**Weaknesses:**

1. According to public information, the SWE-BENCH-VERIFIED index is already very high (78.80%). You should put more experimental results in the experimental section to let everyone confirm the feasibility of your method.

2. There are also some other mainstream benchmarks in the current code field or complex task field. I hope to provide some more experimental results and conclusions, which will make the method more convincing.

3. Your references contain many methods from the past, but few new methods recently. I hope you can add some of the latest methods and provide certain indicator results, which will make your method more recognized.

4 .In your SELECTSOLVE method, the surrogate model seems to play a very important role. I hope to provide more experimental results on the surrogate model to make the method more credible.

**Questions:**

1. Your references only have two works published in 2025. Is there really no other work? In my opinion, solving complex problems has always been important and has attracted much attention.

2.It seems to me that your method is very similar to Retrieval-Augmented Generation. Do you have any relevant literature cited? Or can you explain the difference between your method and Retrieval-Augmented Generation?

3 .The appendix is ​​not well-structured, and it is difficult to give any suggestions.

---

### Official Review · Reviewer_ee2y · 2025-10-31

**Soundness:** 2
**Presentation:** 3
**Contribution:** 3
**Rating:** 4
**Confidence:** 4

**Summary:**

- This paper investigates whether complex agentic scaffolding (multi-step retrieval, tool usage, and iterative reasoning loops) is necessary for solving fully observable tasks such as SWE-Bench. The authors propose a simplified “state-in-context” agent design that places the entire environment state directly into the context window of a Long-Context Language Model (LCLM), allowing it to operate without external tools or structured scaffolding.

**Strengths:**

- Clear and well-motivated research question: The authors identify a meaningful gap—whether agentic complexity is necessary in fully observable environments.

**Weaknesses:**

- Critical Methodological Contradiction Between Core Claims and Actual Implementation. The paper emphasizes that it explores "simply putting the entire environment into the context of a long context language model (LCLM) and properly prompting the model" (Abstract, line 17), while criticizing existing work for requiring "a careful design of agentic scaffoldings tailored to specific tasks" (line 192). However, the actual implementation still follows the agentic workflow approach:
    - In "3.2 State Compression": The method uses "a straightforward ranking-and-filtering approach”(line 262) to filter files, which is equivalent to code localization in Agentless
    - In "3.2 Patch format and validation": The paper adopts identical operations from Agentless, including the Search/Replace edit format, reproduction tests, regression tests, and majority voting
    - In "4.3 ABLATION STUDY" and "A.1 ADDITIONAL RESULTS": These sections explain the impacts of prompting techniques, compression, target file location, and other operations, which contradicts the claim of "simply putting the entire environment"
  - Essentially, the authors still rely on human-engineered agentic workflows to simplify the environment, contradicting the claim that this approach is "simpler to train end-to-end" (line 77). If these operations are intended to mitigate the insufficient capabilities of current LCLMs, it would be more appropriate to validate the approach starting from a task with a smaller environment size.

**Questions:**

- several key analyses are missing:
    - Both LCLM and Agentless "prompt the LM to perform localization based on the compressed repository." Does this mean that Agentless's additional "embedding-based retrieval to localize" actually produces worse localization results? This seems counterintuitive and requires explanation.
    - Combined with the analysis at line 437, does the LCLM's advantage over Agentless stem solely from smaller file-level localization errors?
    - Critical missing experiment: If we control for the same recall (i.e., ensure both methods retrieve the same set of relevant files), does LCLM still maintain its advantage? This would reveal whether the benefit comes from better localization or better utilization of the localized files.

- Others
  - The claim that "multi-hop QA, software engineering do not inherently require active information gathering" is inaccurate. However, many multi-hop QA tasks explicitly require search (e.g., BrowseComp, WebWalker, knowledge base reasoning).
  - Line 177 states "We instantiate state-in-context agents in two ways" but only describes DIRECTSOLVE.
  - Incomplete Prompt Documentation in Appendix C.

---

### Official Review · Reviewer_ckh8 · 2025-11-01

**Soundness:** 3
**Presentation:** 3
**Contribution:** 3
**Rating:** 6
**Confidence:** 4

**Summary:**

This paper investigates whether the complex, multi-step agentic scaffolding currently used for software engineering tasks is necessary. The authors propose a simpler "state-in-context" approach, leveraging Long-Context Language Models to process the entire (or a compressed version of) a code repository in a single pass. They test this on the SWE-Bench-Verified benchmark and introduce two methods: DIRECTSOLVE, where a single LCLM call generates the solution patch , and SELECTSOLVE, a two-stage hybrid where an LCLM first localizes relevant files, which are then fed to a (potentially different) Short-Context LM for repair. The results demonstrate that these simplified, "monolithic" designs are highly competitive.

**Strengths:**

- The paper's core research question is simple, timely, and contrarian. I think this state-in-context design is a useful study case for the agent design space.
- The problem is well motivated. The paper spent quite some time and analysis to show that a lot of coding problems have context that can be reasonable reduced to below 2m. This helps build the case that we can have the proposed state in context.
- The paper's findings are meaningful. It provides a powerful, simple baseline that, with a strong model like Gemini-2.5-Pro, achieves a state-of-the-art 50.8% pass@1 rate. Although I would love to see this compared to some of the more recent scaffolds like AIDE or R&D. But this is just a suggestion. Lack of those doesn't diminish this paper's contribution.

**Weaknesses:**

```Long context performance is a bit contradictory```

Figure 4 and Table 5 show that LCLM performance decreases as context length grows and is highly sensitive to the position of the target file ("lost in the middle"). This strongly suggests that current LCLMs are not effective at "in-context retrieval" over very long, noisy inputs. This finding, which is central to the paper's thesis, should be in the main body. It weakens the "monolithic" DIRECTSOLVE argument and implies that a smarter selection/retrieval step (like that in SELECTSOLVE or RAG) is still fundamentally necessary.

In addition, I foudn the scaffolding-free terminology inaccurate. The SELECTSOLVE is a scaffold abeit being simpler.

```Putting entire environment into context is not entirely true```

The paper's premise of "putting the entire environment into the context"  is not fully met. The data shows that only 21.2% of instances fit this "All Files" definition. The authors get to 85.0% coverage only by aggressively pruning the state to Core Code Only. The compression step itself, which uses an LCLM to rank files based only on the repository structure and issue, is a form of light scaffolding and a potential weak point.

```Cost and Practicality```

The authors rightly acknowledge the high financial cost of the LCLM approach (2.60 per instance vs. 0.25 for Agentless). Their counter-argument relies on KV caching for "repeated queries". However, this is a significant caveat. In a real-world CI/CD pipeline or for a developer working on a new bug, the first query on a given codebase (or a new version of it) will always be a "cold start," incurring the full, high cost. Another counter arguments from the author is that cost will continue to go down. While this is true, and it might even go down in order of magnitudes, the inference-time may also scale in orders of magnitudes. It is always better to do a task with less tokens given that consideration.

**Questions:**

- The cost counter-argument hinges on KV caching for "repeated queries". In a real-world software engineering context where codebases are constantly evolving, how often do you envision this cache being useful? Wouldn't a new commit or branch invalidate the entire cache?
- Another question that's similar to the point above is when a long chain-of-thought or trial and error needs to be done (e.g., in a multi-turn data scientist env), what if what's generated so far exceeds the context length. How is that handled?

---

### Meta-Review · Area_Chair_51tB · 2025-12-28

**Summary:**

This paper investigates whether complex agentic scaffolding (multi-step retrieval, tool usage, and iterative reasoning loops) is necessary for solving fully observable tasks such as SWE-Bench. The authors propose a simplified “state-in-context” agent design that places the entire environment state directly into the context window of a Long-Context Language Model (LCLM), allowing it to operate without external tools or structured scaffolding.

**Reviewer Concerns:**

Strengths:
1. The paper's core research question is simple, timely, and practical. This state-in-context design is a useful study case for the agent design space.
2. The work identifies simple techniques boost real-world system performance, offering practical value to applied practitioners.
3. The paper shows strong empirical results, rivaling or surpassing heavily engineered baselines. It provides a powerful, simple baseline that, with a strong model like Gemini-2.5-Pro, achieves a state-of-the-art 50.8% pass@1 rate.

Weaknesses:
1. Long context performance is a bit contradictory. There is critical methodological contradiction between core claims and actual implementation.
2. Putting entire environment into context is not entirely true.
3. Cost and Practicality is not properly discussed.
4. Evaluation, experimental results, and analysis are not complete, more datasets or tasks are necessary.

**Reviewer Scores:**

the ratings come out as 2,4,4,6. no author responses were provided.

---

### Decision · Program_Chairs · 2026-01-26

Reject